# Piezo1 and Its Function in Different Blood Cell Lineages

**DOI:** 10.3390/cells13060482

**Published:** 2024-03-09

**Authors:** Anastasia Iris Karkempetzaki, Katya Ravid

**Affiliations:** 1Department of Medicine, Chobanian & Avedisian School of Medicine, Boston University, Boston, MA 02118, USA; aikarke@bu.edu; 2Whitaker Cardiovascular Institute, Chobanian & Avedisian School of Medicine, Boston University, Boston, MA 02118, USA; 3School of Medicine, University of Crete, 71003 Heraklion, Greece

**Keywords:** ion channels, Piezo1, blood cells, megakaryocytes, platelets

## Abstract

Mechanosensation is a fundamental function through which cells sense mechanical stimuli by initiating intracellular ion currents. Ion channels play a pivotal role in this process by orchestrating a cascade of events leading to the activation of downstream signaling pathways in response to particular stimuli. Piezo1 is a cation channel that reacts with Ca^2+^ influx in response to pressure sensation evoked by tension on the cell lipid membrane, originating from cell–cell, cell–matrix, or hydrostatic pressure forces, such as laminar flow and shear stress. The application of such forces takes place in normal physiological processes of the cell, but also in the context of different diseases, where microenvironment stiffness or excessive/irregular hydrostatic pressure dysregulates the normal expression and/or activation of Piezo1. Since Piezo1 is expressed in several blood cell lineages and mutations of the channel have been associated with blood cell disorders, studies have focused on its role in the development and function of blood cells. Here, we review the function of Piezo1 in different blood cell lineages and related diseases, with a focus on megakaryocytes and platelets.

## 1. Overview of Piezo1 Properties

### 1.1. Discovery of Piezo Proteins and Structure of Piezo1

Research on mechanotransduction dates back to the mid-1800s, prompted by awareness of skin mechanosensation [1]. Yet, it was not until the 1980s that evidence of ion channel acquisition and conversion of force to electrical currents was demonstrated with the availability of the patch-clamp technique [2,3,4]. The next significant step in the field came around the 2000s with the identification of transient receptor potential (TRP) channels that mediate sensations of temperature and chemical irritants [5,6]. Experiments on sensory neurons demonstrated that their ion channels evoke currents much larger and more distinct than those evoked by the ion channels of non-sensory neurons, which led to the conclusion that mechanosensory ion channels play an important role and have important properties, leading to investigations on their mechanisms of action [7,8]. A significant milestone in the identification of sensory mechanisms was achieved with the discovery of the touch receptor, by screening a neural crest cell-derived cancer cell line (N2a cells) [9]. In this study, Coste et al. first identified candidate molecules based on results from genetic approaches and pressure-clamp responses. Loss-of-function knockdown experiments identified Family With Sequence Similarity 38 genes, termed *Fam38a*, conserved among eukaryotic species. The family was identified by a neuroscientist, Ardem Patapoutian (earning him a Nobel prize in physiology in 2021), who named the two related ion channel proteins Piezo1 (Fam38a) and Piezo2 (Fam38b)—from the Greek word *piezein*, meaning squeeze/press. Piezo1 and Piezo2 mobilize calcium into cells in response to mechanical forces [9,10,11].

Piezo channels act as mechanosensors, sensing tension in the plasma membrane [12] and, thus, mediating the sensation of touch and pain, proprioception, balance, hearing, blood flow and other fundamental biological processes, such as blood pressure regulation and cell migration and proliferation [13,14]. Functioning through a process called mechanotransduction, Piezo channels convert mechanical stimuli into electrochemical signals via allowance of Ca^2+^ influx across the plasma membrane. Because of their function, studies have focused on the channels’ mechanism of action and structure. Firstly, the topology of the channels on the plasma membrane and the importance of the C-terminal domain as the ion permeation pathway were characterized [15,16,17]. Using cryo-electron microscopy, different studies revealed the structure of Piezo1 channel as a trimer consisting of three identical Piezo1 monomers, exhibiting a three-blade propeller shape. Each subunit consists of 38 transmembrane domains creating repetitive transmembrane helical units. The central ion-conducting channel or pore is formed by the two transmembrane helices closest to the C terminus of each subunit, creating local deformation of the surrounding membrane [18,19,20,21]. The components participating in transduction are the distinct structures of the beam, anchor and C-terminal domain, outer helix and inner helix, which functionally connect to fulfill a lever-like mechanogating of Piezo1, as diagrammed in other reports [22,23].

### 1.2. Piezo1 Expression and Cellular Distribution

While Piezo1 is expressed in different cell types, such as endothelial cells; cardiomyocytes; blood cells; cancer cells; stem cells; and progenitors such as hematopoietic stem cells, osteoblasts, adipocytes and astrocytes, its distribution in the cells is dynamic and varies from cell to cell. Pieoz1 molecules are enriched in subcellular areas where mechanotransduction is important for functions such as focal adhesion [24], wound healing [25] and cytokinesis [26]. Piezi1 is found in t-tubules within cardiomyocytes [27], in biconcave “dimples” in red blood cells [28], in intracellular cytoskeletal bridges during cytokinesis [26] and in the rear of migrating keratinocytes during wound healing [25]. A recent study demonstrates that membrane curvature is a key regulator of Piezo1 distribution [29]. In this study, it was shown that Piezo1 is depleted from membrane protrusions, where membrane curvature is negative, while it is more abundant in membrane invaginations where the membrane curvature is positive [29]. This is explained by Piezo1 being flattened in its active form [30,31]. In this way, it can promote endocytosis by driving membrane invagination formation, and it inhibits the formation of protrusions such as filopodia [29]. Apart from the membrane curvature, membrane tension plays an important role in Piezo1 distribution as it can change the channel’s nano-geometries [32]. Recently, it was found that Piezo1, like Piezo2, localizes at centrosomes, playing an important role in cell cycle progression by regulating centrosome integrity. The dysregulation of this channel expression results in the formation of supernumerary centrosomes, which is one of the hallmarks of cancer [33].

Upregulation of the channel can happen through different stimuli such as shear stress; increased stiffness of the extracellular environment; and signals coming from injury, inflammatory cytokines or other metabolites. Shear stress is implicated in Piezo1 upregulation in various cell types such as in endothelial and vascular cells in the context of pulmonary hypertension [34,35], in endothelial cells via adrenomedullin [36] and in megakaryocytes (MKs) and platelets with the activation of integrins [37]. Stiffness sensation by Piezo1 was found to be associated with upregulation of the expression of the channel in MKs, as proven by culturing these cells in a 3D culture medium of different stiffness and by its enhanced MK expression in the case of primary myelofibrosis (PMF), a disease characterized by bone marrow fibrosis and augmented stiffness of the extracellular matrix (ECM) [38]. Stiffness created by kidney fibrosis was also shown to cause Piezo1 upregulation and to be associated with the progression of the disease [39]. Stiffness caused by extracellular amyloid-β (Aβ) plaques formed in Alzheimer’s disease (AD) leads to Piezo1 upregulation in microglia, resulting in activation of these cells to combat the disease [40]. Piezo1 upregulation in astrocytes dampens neuroinflammation caused in AD brains by inhibiting the release of cytokines and chemokines [41]. Similarly, Piezo1 is upregulated within an inflammatory milieu in a pre-clinical model of mesio-temporal lobe epilepsy, and, accordingly, TNFα upregulated Piezo1 expression in astrocytes [42]. Piezo1 expression is also augmented in various cancers due to stiffness in the tumor microenvironment, causing further cell proliferation and migration [43]. Moreover, injury-mediated Piezo1 upregulation has been reported in the cases of peripheral nerve injury [44] and bone marrow irradiation injury [45]. Finally, metabolites such as Trimethylamine-N-oxide, a diet-derived metabolite that correlates positively with multiple chronic diseases, were shown to upregulate Piezo1 in chondrocytes and to correlate with the pathogenesis of osteoarthritis [46].

### 1.3. Mechanogating of Piezo1 and Lipid Interactome

A link between membrane curvature, clustering and Piezo1 function was proposed. Interestingly, Piezo1 might be activated in the absence of cytoskeletal forces, as evidenced by the lack of influence of actin-depolymerization toxin treatment on Piezo1 activation [47]. Borbiro et al. showed that Piezo1 and Piezo2 inhibition is potentiated by the depletion of plasma membrane phosphoinositides in response to phospholipase C (PLC) activation by Ca^2+^ influx via the Transient Receptor Potential Vanilloid 1 (TRPV1) channel. The effect of excess phosphatidylinositol biphosphates (PI(4,5)P2) or PI(4)P addition was also investigated; it delayed, rather than eliminated, Piezo1 current inhibition by TRPV1 activation, suggesting a complex involvement of other signaling pathways and not a direct activation of Piezo1 from lipids [48,49]. This highlights the role of lipids and the membrane environment in controlling Piezo1 activity, although it does not exclude an enhancing effect of cytoskeletal components on Piezo1 activity [50,51]. An element of the lipid bilayer shown to affect the gating of the channel is cholesterol content, which influences the physical properties of the membrane and, thus, transmembrane protein properties. Also, negatively charged lipids such as PIP2 [52] or lipids with different fatty acid saturation levels [47] can directly affect this protein’s activity through changes in the stiffness and rigidity properties of the membrane [53,54]. The lipid “fingerprint” was studied with membrane stimulations of different lipid contents, where cholesterol and PIP2 binding sites of Piezo1 were identified, and the importance of these molecules for activation of the channel was highlighted [55,56]. Apart from the lipid membrane constitution itself, the curvature of the membrane is very important for the activation of Piezo1 and affects the clustering of the channel. While the channel is closed, it deforms the local membrane into a cup shape. Upon mechanical stimulation, the channel flattens the membrane with the propeller blades straightening, providing the energy to open the channel [21,30]. It is still controversial how clustering affects activation. Despite the idea that open channels have a less curved membrane around them, thus reducing the energy of association and allowing aggregation, enhanced curvature can lead to the clustering of other channels [57,58]. The above studies propose a force-from-lipid model [59] for Piezo channel activation, suggesting that channel activation can happen without the influence of the cytoskeleton and the extracellular matrix. The alternative model proposed is the force-from-filament model, which has been shown before to be an ion channel activation mechanism [60,61].

### 1.4. Piezo1 Intracellular Signaling and Associated Proteins

Piezo1 activation leads to an array of intracellular signaling in different cell types. Downstream Piezo1-mediated Ca^2+^ signaling and activation of different kinases can lead to different cascades of events (Figure 1) promoting or inhibiting various cellular processes. Focal adhesion kinase (FAK) activation depends on the sustainability of intracellular Ca^2+^ influx, which can lead to a bimodal pattern of FAK response, mediated by the Src homology region 2 (SH2)-containing protein tyrosine phosphatase 2 (SHP2) [62]. Interestingly, different flow patterns activate the same initial Piezo1-G_q_/G_11_-mediated signaling. Downstream of this signaling, integrins are activated, leading to focal FAK and nuclear factor kappa-light-chain-enhancer of activated B cells (NF-κB) activation and promotion of inflammation. This occurs in the case of disrupted flow. On the other hand, in the case of laminar flow, integrins are not activated, leading to anti-inflammatory signaling medicated by phosphoinositide 3-kinase (PI-3)-kinase and AKT in endothelial cells [63]. Also in endothelial cells, Piezo1 regulates the rho-associated coiled-coil forming kinases (ROCK1/2) pathway by decreasing its phosphorylation and consequently reducing the expression of claudin-1, a key protein in tight junctions of endothelial cells, leading to endothelial cell dysfunction [64]. In cardiac fibroblasts, Piezo1 activation leads to elevated IL-6 expression and secretion associated with p38/mitogen activated protein kinase (MAPK) signaling, thus impacting cardiac remodeling [65]. The deletion of Piezo1 specifically in mouse cardiomyocytes prevented the activation of calmodulin-dependent kinase II and inhibited a hypertrophic response to pressure overload [66]. In mesenchymal stem cells, activation of Piezo1 stimulates these cells’ migration by inducing ATP release and subsequent activation of P2 purinergic receptor signaling, which further activates proline-rich tyrosine kinase 2 (PYK2) and MAK/extracellular-signal-regulated kinase (ERK) signaling pathways [67].

Apart from the activation of kinases, Piezo1 activation affects the phosphorylation and cellular localization of the transcription factors yes-associated protein (YAP) and transcriptional co-activator with PDZ-binding motif (TAZ). In neuronal stem cells, the inhibition of Piezo1 reduces the nuclear localization of YAP, resulting in the suppression of neurogenesis [68]. Hippo pathway inactivation and nuclear translocation of YAP/TAZ are also initiated upon Piezo1 activation, leading to increased cell proliferation in oral squamous cell carcinoma, facilitated by a stiff ECM microenvironment of the tumor [69]. Another study highlighted the contribution of Piezo1 to tumorigenesis in hepatocytes through the activation of MAPK signaling mediated by the phosphorylation of Jun N-terminal kinase (JNK), p38 and ERK, leading to downstream nuclear translocation of YAP as an essential pathway for cells’ function and proliferation [70]. The contribution of Piezo1 activation to the control of other transcription factors was also reported in the context of bone formation where Piezo1-induced Jun N-terminal kinase (NFAT)/YAP1/ß-catenin signaling promotes osteogenesis [71]. In endothelial cells, Piezo1 signaling leads to the activation of Krüppel-like factors (KLFs) and regulation of cellular homeostasis [72]. Interestingly, transcription factors can regulate Piezo1. A recent study identified the MyoD (myoblast determination)-family inhibitor proteins (MDFIC and MDFI) to be Piezo1/2 auxiliary subunits in human dermal fibroblasts. These transcriptional regulators directly bind Piezo1/2 and regulate their inactivation kinetics upon their lipidation [73].

Piezo activation leads to integrin activation through an inside-out signaling mechanism mediated by Ca^2+^-dependent PKC–calpain pathway activation in erythroblasts [74]. Similar integrin activation via a Piezo1–calpain axis was also shown in the HK2 kidney cell line, mediating the adhesion of the cells and the progression of fibrosis, while activation of TGF-ß was initiated [39]. Direct interaction of Piezo1 with adhesion molecules was also reported. PECAM-1 and CDH5 interact with Piezo1 and direct it to cell–cell junctions in endothelial cells [75]. Piezo1 is an integrin-interacting protein. A role of Fam38a (Piezo1) in cell adhesion through integrin ß1 was reported in epithelial cells, showing localization at the endoplasmic reticulum, Ca^2+^ release and R-Ras recruitment as a mechanism of action [76]. Recently, Piezo1 was also identified as an integrin-interacting protein in focal adhesions of normal fibroblasts. Specifically, upon cell adhesion to fibronectin, Piezo1 is recruited to integrin ß3 adhesion sites where it remains, while integrin ß3 is dissociated and integrin ß1 takes its place and colocalizes with Piezo1. As a result, excessive Ca^2+^ entry and recruitment of calpain hydrolyze components of the adhesion complex, causing disassembly of adhesion and thereby regulating cell growth and migration. This adhesion process is absent in transformed cancer cells, where growth appears independent of local Ca^2+^ entry at rigid matrix sites, and Piezo1 is decoupled from traction forces and adhesion signaling [77].

### 1.5. Pharmacological Activators and Inhibitors of Piezo1

Studies on Piezo1 have involved the use of pharmacological modulators, including the small molecule 2-[(2,6-Dichlorophenyl)methylsulfanyl]-5-pyrazin-2-yl-1,3,4-thiadiazole (Yoda-1), a chemical agonist known to activate Piezo1, and Grammastola spatulata mechanotoxin-4 (GsMtx4), a specific antagonist of the channel. These useful tools have been employed to study Piezo1’s function in different cells and animal models, in the context of normal physiological processes or in disease. Yoda-1 was identified with a high-throughput screening of over 3 million low-molecular-weight compounds in HEK cells transfected with Piezo1/2 cDNA. In the absence of mechanical forces, Piezo1 can be activated by Yoda-1, which stabilizes the channel in its open state and slows down its inactivation kinetics [78]. A recent study introduced a new modification in Yoda-1 with the substitution of pyrazine with 4-benzoic acid, which proved to have improved PIEZO1 agonist activity [79]. Another attempt at Yoda-1 pyrazine ring modification led to the identification of an analog, Dooku1, that did not have agonist ability but could reversibly antagonize Ca^2+^ entry induced by Yoda-1 [80]. The specific inhibition of the channel can be induced with the toxin peptide GsMTx-4, which binds to the membrane near the channel, leading to partial relaxation of the outer membrane and finally inhibiting the effective activation of the channel. GsMTx4 is a small (3–5 kD) amphipathic molecule built on a conserved inhibitory cysteine knot (ICK) backbone which is unique due to its highly positive charge (+5) coming from its six lysine residues. This characteristic qualifies GsMTx4 as a specific Piezo inhibitor as it leads to tension-dependent depth changes of GsMTx4 that modulate the area of the outer monolayer of the cell [81].

## 2. Piezo1 Function in Blood Cell Lineages

### 2.1. Piezo1 in Megakaryocytes and Platelets

Using human platelets and the megakaryocytic cell line Meg-01, Ilkan et al. showed that shear stress induces a sustained increase in intracellular Ca^2+^. Inhibiting platelet Piezo1 with the GsMtx4 inhibitor ameliorated the Ca^2+^ increase, while thrombus size was reduced by 50% in vitro. Moreover, the Piezo1 agonist Yoda-1 seemed to induce Ca^2+^ influx by 170%, indicating that this channel is implicated in shear stress responses and Ca^2+^ influx in human platelets and Meg-01 cells [37]. A recent study using primary bone marrow cells showed that Piezo1 acts as a brake for MK maturation and platelet production in vitro, as well as an attenuator of these processes in vivo in mice, as shown by the deletion of Piezo1/2 in the megakaryocytic lineage [38]. Elevated expression of the Piezo1 protein was observed in MKs isolated from the bone marrow of mice or patients affected by JAK2V617F-induced primary myelofibrosis, a clonal hematological neoplasia characterized by a stiff ECM in the bone marrow microenvironment [38].

Other studies have elucidated a role of Piezo1 in platelet activation, building on the hypothesis that mechanosensation plays a pivotal role in thrombosis through platelets sensing external mechanical forces, leading to aggregation and thrombus formation [37,82]. Experiments involving in vivo injections of the Piezo1 agonist Yoda1 or antagonist GsMtx4 in a mouse model of hypertension led to the conclusion that Piezo1 activation in platelets causes platelet hyperactivation and increases the risk of thrombosis. Based on mechanistic studies, it was proposed that Piezo1 is associated with mitochondrial dysfunction and apoptosis in platelets, a process modulated by the PI3K/AKT signaling pathway. Conversely, inhibition of Piezo1 rescued hypertensive mice from thrombosis and stroke [82].

### 2.2. Piezo1 and Red Blood Cells

Piezo1’s role in red blood cell (RBC) biology has been well characterized through studies investigating mechanisms of disease. Patients with dehydrated hereditary somatocytosis (DHS) manifesting hemolytic anemia have been shown to carry gain-of-function mutations in the *Piezo1* gene (DHS1; 194380), correlated with altered RBC mechanical responses and increased Ca^2+^ cellular influx [83,84]. Studying DHS led to the conclusion that Piezo1 is associated with the regulation of RBC volume. Specifically, gain-of-function mutations in Piezo1 in patients with DHS have been associated with dehydration of RBCs [84]. In accordance with this observation, Piezo1 activation by Yoda-1 resulted in Ca^2+^ influx and dehydration of RBCs via increased Ca^2+^ influx and activation of K^+^ efflux through the Gardos channel (KCa3.1), leading to water loss and RBC dehydration. In contrast, the generation of RBC-specific Piezo1 conditional knockout mice demonstrated an overhydrated RBC phenotype [28,84]. A recent study identified Piezo1-generated intracellular Ca^2+^ as an activator of the phospholipid scramblase CaPLSase in RBCs, which is enhanced in patients with a hereditary gain-of-function mutation in *Piezo1*, manifested in xerocytosis [85]. Piezo1 is also associated with sickle cell disease through the Psickle pathway that regulates erythrocyte volume and is partially responsible for the sickling of RBCs together, with the involvement of other ion channel systems, such as the K-Cl cotransporters and the Gardos channel. The observed increased cation permeability in diseased cells was posited to be mediated via a stretch-activated channel, with the most possible candidate being Piezo1. A further indication of this mechanism is the inhibitory effect of GsMTx-4 on the Psickle pathway [86]. In addition, the observation that DHS patients have an iron overload phenotype prompted investigations on the role of Piezo1 mechanotransduction in iron metabolism. Macrophage-specific gain-of-function Piezo1 mice had increased iron deposition, transferrin saturation and ferritin concentration, and the macrophages in these mice mediated iron homeostasis in a Piezo1-dependent manner [87].

Recent studies have shown that Piezo1 not only regulates erythrocyte volume, but also regulates a mechanotransductive ATP release from RBCs by controlling shear-induced Ca^2+^ influx in the cells [88]. Computational studies sought to determine the role of Piezo1 in RBC homeostasis during circulatory aging of RBCs. Piezo1-mediated Ca^2+^ permeation above a certain point in the circulation was found to be associated with the development of a pattern of early or late hyperdense collapse, referring to an extreme dehydration response of RBCs, followed by delayed density reversal, referring to the densities of RBCs with different hemoglobin contents in the blood during their lifespan cycle [89,90]. Finally, Piezo1 was also proposed to have a role in controlling Er antigen expression in erythrocytes. Studies of individuals with alloantibodies against the Er antigen associated with a missense mutation in the Piezo1 gene led to the discovery that Piezo1 is required for Er antigen expression and established Er as a new blood group system [91].

### 2.3. Piezo1 and Myeloid Cells

Myeloid cells originate from hematopoietic stem cells (HSCs) that give rise to common myeloid progenitors (CMPs) and in turn divide into monocytes and granulocytes. Monocytes further differentiate into macrophages in various tissues or dendritic cells (DCs) in lymphoid organs [92]. Here, we will discuss the role of Piezo1 in monocytes and the different myeloid cells.

Piezo1 plays a role in macrophage inflammatory responses. Mechanotransduction through Piezo1 induces a metabolic switch towards aerobic glycolysis in macrophages via the Ca^2+^-induced CaMKII-HIF1a axis, which can in turn potentiate inflammatory responses [93]. Piezo1 was also shown to regulate macrophage polarization and activation via an actin–Piezo1 positive feedback mechanism. This results in changes in the macrophages’ inflammatory and healing responses by polarization towards either the M1 proinflammatory or the M2 anti-inflammatory phenotype, in stiffer or softer substrates, respectively [94]. In addition, cross-talks between integrins and Piezo1 were observed in the process of macrophage activation [95]. In this study inflammatory responses from activated macrophages with increased expression of CD11b were suppressed by Piezo1-mediated stretch sensation via an actin-dependent pathway. Furthermore, the role of Piezo1 is extended to its action as a co-receptor of TLR4 in macrophages, leading to enhanced bacterial clearance through phagocytosis and activation of innate responses. Deficiency of Piezo1 in bone-marrow-derived macrophages (BMDMs) results in impaired cytoskeletal remodeling, F-actin organization and ROS production, with a proposed mechanism of action through the CaMKII-Mst1/2-Rac axis [96].

The conditional knockout of Piezo1 from myeloid cells showed that infiltrating monocytes responding to mechanotransduction via Piezo1 recruit neutrophils to clear infections [97]. Additional roles of Piezo1 in immunity and cancer were revealed by a study where Piezo1 deletion from myeloid cells in a polymicrobial sepsis model was shown to be related to less death, enhanced bacterial clearance and decreased proinflammatory cytokine release. In pancreatic ductal adenocarcinoma (PDA) mouse models, Piezo1 deletion from myeloid cells was found to result in enhanced intratumoral CD4+ and CD8+ T-cell activation and protection against cancer progression [98].

Tension and microenvironment stiffness are also associated with alterations in the function and metabolism of dendritic cells (DCs). DCs are antigen-presenting cells (APCs) with an important role in initiating inflammatory responses and anti-tumor immunity. Interestingly, Piezo1 seems to be a link between metabolism and DC function, as the channel was shown to activate a DC-driven differentiation of TH1 and T regulatory (Treg) cells in cancer in association with changes in glycolysis pathways [99]. While Piezo1 deletion from DCs led to decreased anti-tumor immune response due to the inhibition of TH1 generation and an increase in the production of Treg cells [99], Piezo1 activation by mechanical stiffness induced proinflammatory responses from DCs via activation of the downstream Hippo pathway [100].

### 2.4. Piezo1 and Immune Cells

Apart from myeloid cells, CD4+ and CD8+ T cells also express Piezo1 with an important role in T-cell activation. Mechanotransduction in the immunological synapse of T-cell receptors (TCRs) and peptides presented by the major histocompatibility complex molecules (pMHCs) is essential for the optimization of TCR activation [101]. Following TCR-APC assembly, multiple mechanical forces activate Piezo1 to induce Ca^2+^ influx in the cells, activating the downstream calpain pathway, which results in actin cytoskeleton reorganization and stabilization of the immunological synapse. Lymphocyte function-associated antigen 1 (LFA-1) is an integrin receptor activating actin polymerization following TCR-APC adhesion and in parallel also enhancing sensitivity to mechanical forces in a positive feedback loop mechanism [102]. Piezo1 may also play a role in initially activating LFA-1, as it was shown to activate integrins in an inside-out mechanism in other cells [74], and recruiting R-RAS GTPases to the ER, causing Ca^2+^ release [76].

Piezo1 was also proposed as a therapeutic target, indicated by a novel system of remote and precise activation of chimeric antigen receptor (CAR) T cells against solid tumor cells with the use of high-frequency ultrasound stimulation of Piezo1. Such stimulated Piezo1 allows the influx of Ca^2+^, enabling translocation to the nucleus of calcineurin-dependent NFAT and driving the downstream expression of tumor-specific CARs [103]. Another study showed that focused ultrasounds activate mechanosensation in tumor cells, leading to enhanced antigen presentation and, thus, enhanced T-cell activation overcoming tumor-induced immunosuppression [104].

Finally, Piezo1 has a role in autoimmunity too. Using a mouse model of autoimmune encephalomyelitis (EAE), it was shown that Piezo1 deletion minimizes disease severity while expanding the Treg pool via increased TGF-ß signaling. Additionally, knocking out Piezo1 in CD4+ T cells showed no disruption of the normal T-cell differentiation into TH1 and TH17 effector cells, proliferation and homing in the lymph nodes [105].

## 3. Discussion and Future Directions

Mechanically activated ion channels have important roles in essential cell processes, with Piezo1 being one of the important channels involved in blood cell development and functions, as summarized in Table 1. Mutation of *Piezo1* leads to severe diseases and disruption of homeostasis. Studies of patients with hereditary somatocytosis have mainly focused on the red blood cells. Future studies could examine the levels, function and volume of platelets in patients with different Piezo1 mutations, considering findings in mice with the deletion of Piezo1. For instance, a new study highlighting roles for Piezo1 in controlling megakaryocyte development and platelet levels in health and under JAK2 mutation-induced myeloproliferative disease suggests its potential as a therapeutic target in this pathology. Such studies also open the inquiry of whether other mutations that induce clonal hematopoiesis and change bone marrow stiffness have an augmenting effect on Piezo1 expression, Other studies linking Piezo1 activation to enhanced CAR T-cell therapy against solid tumor cells highlight the need for greater translational studies to develop and explore the potential of new Piezo1 modulators as therapeutic agents. The interplay between Piezo1 and integrins and other adhesion molecules is of high importance for Piezo1’s activation and downstream signaling events that coordinate physiological processes. Ongoing studies on the mechanisms of action in each pathway in specific cells could also enhance Piezo1’s significance as a key cellular modulator in different diseases. Finally, investigating overlapping and distinct properties of Piezo1 and Piezo2 in health and different disease states will enhance knowledge related to possibly common therapeutic targets.

## Figures and Tables

**Figure 1 cells-13-00482-f001:**
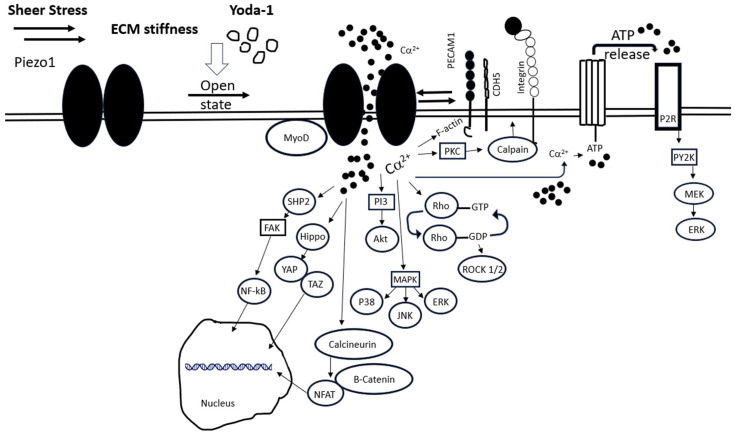
Illustration of main signaling pathways triggered by Piezo1 activation in various cell types. Illustrated here are major signaling pathways known to be activated by Piezo1. However, as detailed in the text, Piezo1-triggered signaling pathways could differ in different cells, resulting in different cellular functions. These properties depend on the cell type and, naturally, might be altered upon cell exposure to varying external forces and conditions. The abbreviations of depicted proteins are denoted in the text.

**Table 1 cells-13-00482-t001:** Highlights of major Piezo1 functions in cellular processes in the different blood cell types.

Cell Type	Piezo1 Function	References
Megakaryocytes (MKs)	MK development	[38]
	MK shear stress responses	[37]
Platelets	Platelet biogenesis	[38]
	Platelet activation	[37,82]
	Platelet shear stress responses	[37]
Red blood cells (RBCs)	RBC volume regulation	[83,84]
	Iron metabolism	[87]
	Erythropoiesis	[74]
	Er antigen expression	[91]
	RBC ATP release	[88]
	RBC circulatory aging homeostasis	[89,90]
Macrophages	Metabolic switch to anaerobic glycolysis/inflammatory responses	[93]
	Macrophage activation/phagocytosis (TLR-4 co-receptor)	[95,96]
Dendritic cells (DCs)	DC-mediated differentiation of TH1 and Treg cells	[99]
	DC function (Ag presentation) and metabolism control	[100]
T cells	Stabilization of the immunological synapse (TCR-APC cell)	[101]
	T-cell activation in cancer and autoimmunity	[104,105]

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
