# Peer review of "Piezo1 and Its Function in Different Blood Cell Lineages"

_cells, 2024, doi:10.3390/cells13060482_

Round 1
Reviewer 1 Report
Comments and Suggestions for Authors
This is a very complete and well-written review of Piezo 1 biology and function in different blood cell lineages. It will be a very nice addition to the field. I only have a few suggestions to improve the manuscript.
1) Can the authors state whether human genome studies have shown an association between Piezo1 SNPs and RBC or platelet volume?
2) The review would benefit from a diagram of the Piezo 1 structure to complement the description on page 2 of the manuscript.
3) On page 5, Line 199, the authors mention an “adhesion maturation process”. Can they clarify what this means?
4) The description of the studies demonstrating the association of Piezo1 with the regulation of RBC volume on page 6, lines 250-254, are unclear. It sounds like both the RBC specific Piezo1 conditional KO mice and the mice treated with the Piezo1 activator Yoda-1 have a dehydrated RBC phenotype. Can the authors clarify this?
Author Response
This is a very complete and well-written review of Piezo 1 biology and function in different blood cell lineages. It will be a very nice addition to the field. I only have a few suggestions to improve the manuscript.
Overall response: We thank the reviewer for their overall positive evaluation and constructive comments, which we have fully addressed. Revised texts are denoted in yellow highlights.
1) Can the authors state whether human genome studies have shown an association between Piezo1 SNPs and RBC or platelet volume?
Response 1: This is an excellent point. There are no published data on this point. Interestingly, Dr. Seth Alper at Beth Israel Deaconess Medical Center diagnosed three patients with hereditary xerocytosis arising from heterozygous activating mutations in Piezo1. To our past inquiry, Dr. Alper checked and reported to us mild and severe thrombocytosis in two of the three patients. The revised paper includes this point as a proposed future research direction, as denoted under Discussion and future directions.
2) The review would benefit from a diagram of the Piezo 1 structure to complement the description on page 2 of the manuscript.
Response 2: The complex structure of Piezo1 has been published by others and in other reviews focused in Piezo biophysical properties. Any diagram of its structure will largely duplicate other published ones. We are, therefore, directing the reader to other papers cited as references #18-23.
3) On page 5, Line 199, the authors mention an “adhesion maturation process”. Can they clarify what this means?
Response 3: We thank the reviewer for noticing. The word maturation was removed.
4) The description of the studies demonstrating the association of Piezo1 with the regulation of RBC volume on page 6, lines 250-254, are unclear. It sounds like both the RBC specific Piezo1 conditional KO mice and the mice treated with the Piezo1 activator Yoda-1 have a dehydrated RBC phenotype. Can the authors clarify this?
Response 4: We regret this unintended confusion. The point has been clarified (see revised text in lines 211-215)
Reviewer 2 Report
Comments and Suggestions for Authors
This review manuscript is aiming to summarize Piezo1 channel and Its Function in Different Blood Cell Lineages. It is providing an interesting topic and in general, it is well structured. But there are still a few weaknesses spotted.
Specific comments:
1. In line 155: ‘In myocardiocytes’ arising from reference 65 should be ‘In cardiac fibroblasts’. Since the authors are interested in cardiac remodelling, they may include the new findings from cardiomyocytes as reported by Nature Cardiovascular Research 1, 577–591 (2022).
2. Figure 1: the illustration is somehow misleading, as the readers are not sure which pathway downstream Piezo1 activation is responsible for different roles in various types of cells. I would suggest the authors to list each type of cells and illustrate different pathways for each type of cells.
3. The Piezo1 interacting molecules involved in cell adhesion is discussed in this manuscript, but the latest progress on this area is missing: Commun Biol. 2023 Apr 1;6(1):358.
4. Piezo1 activators and inhibitors have been expanded by a few newer papers and these papers should be included: Br J Pharmacol. 2023 Aug;180(16):2039-2063; Br J Pharmacol. 2018 May;175(10):1744-1759.
5. The roles and their underlying pathways in the different blood cell lineages should be illustrated as separate figures.
Author Response
This review manuscript is aiming to summarize Piezo1 channel and Its Function in Different Blood Cell Lineages. It is providing an interesting topic and in general, it is well structured. But there are still a few weaknesses spotted.
Overall response: We appreciate the reviewer finding the topic to be interesting. We also thank the reviewer for their comments, which we have fully addressed. Revised texts are denoted in yellow highlights.
Specific comments:
- In line 155: ‘In myocardiocytes’ arising from reference 65 should be ‘In cardiac fibroblasts’. Since the authors are interested in cardiac remodelling, they may include the new findings from cardiomyocytes as reported by Nature Cardiovascular Research volume 1, pages577–591 (2022).
Response 1: We thank the reviewer for this suggestion. Our review is focused on blood cells. Adding functional information related to cardiomyocytes, although important, would raise questions related to focus and selective inclusion (e.g., why not data related to kidney or other tissues). Yet, the paper cited by the reviewer adds an interesting facet related to signaling. We added the reference in context of signaling pathways (lines 130-131 in the revised paper).
- Figure 1: the illustration is somehow misleading, as the readers are not sure which pathway downstream Piezo1 activation is responsible for different roles in various types of cells. I would suggest the authors to list each type of cells and illustrate different pathways for each type of cells.
Response 2: We understand the reviewer’s concern. We certainly wish to avoid misleading. As the text indicated, indeed, several cell tyles respond to Piezo via different signaling pathways. We considered an image as suggested by the reviewer, but based on early sketching have realized that it would be too cumbersome, and at the end might too be misleading, i.e., not inclusive of all possible signaling reported for different cells, which would be altered based on changing external forces and exposures. To address this, we changed the legend text to our figure to indicate: “Illustrated here are major signaling pathways known to be activated by Piezo1. However, as detailed in the text, Piezo1-triggered signaling pathways could differ in different cells, resulting in different cellular functions. These properties depend on the cell type, and, naturally, might be altered upon cell exposure to varying external forces and conditions.”
- The Piezo1 interacting molecules involved in cell adhesion is discussed in this manuscript, but the latest progress on this area is missing: Commun Biol. 2023 Apr 1;6(1):358.
Response 3: The paper was cited originally as reference # 74. We have now completed the information in the cited paper (now cited as reference # 75).
- Piezo1 activators and inhibitors have been expanded by a few newer papers and these papers should be included: Br J Pharmacol. 2023 Aug;180(16):2039-2063; Br J Pharmacol. 2018 May;175(10):1744-1759.
Response 4: we greatly appreciate these added insights. These papers and brief descriptions were added to the revised texts (lines 170-174).
- The roles and their underlying pathways in the different blood cell lineages should be illustrated as separate figures.
Response 5: Please see our response to comment # 3. Following attempts and guided by a professional illustrator, in this case we resorted to Table 1 that addresses the point raised by the reviewer. It allows clean presentation of effects of Piezo in different blood cell lineages, backed up by cited references.
Round 2
Reviewer 2 Report
Comments and Suggestions for Authors
The authors have addressed my original concerns and questions.